# Tunable molecular separation by nanoporous membranes

Zhengbang Wang[1,*], Alexander Knebel[2,*], Sylvain Grosjean[3], Danny Wagner[4], Stefan Bräse[3,4,5], Christof Wöll[1], Jürgen Caro[2] & Lars Heinke[1]

Metal-organic frameworks offer tremendous potential for efficient separation of molecular mixtures. Different pore sizes and suitable functionalizations of the framework allow for an adjustment of the static selectivity. Here we report membranes which offer dynamic control of the selectivity by remote signals, thus enabling a continuous adjustment of the permeate flux. This is realized by assembling linkers containing photoresponsive azobenzene-side-groups into monolithic, crystalline membranes of metal-organic frameworks. The azobenzene moieties can be switched from the *trans* to the *cis* configuration and *vice versa* by irradiation with ultraviolet or visible light, resulting in a substantial modification of the membrane permeability and separation factor. The precise control of the *cis:trans* azobenzene ratio, for example, by controlled irradiation times or by simultaneous irradiation with ultraviolet and visible light, enables the continuous tuning of the separation. For hydrogen:carbon-dioxide, the separation factor of this smart membrane can be steplessly adjusted between 3 and 8.

[1] Institute of Functional Interfaces (IFG), Karlsruhe Institute of Technology (KIT), Hermann-von-Helmholtz-Platz 1, 76344 Eggenstein-Leopoldshafen, Germany. [2] Institute for Physical Chemistry and Electrochemistry, Leibniz University Hanover, Callinstraße 3a, 30167 Hannover, Germany. [3] Soft Matter Synthesis Lab, Institute of Biological Interfaces 3 (IBG 3), Karlsruhe Institute of Technology (KIT), Hermann-von-Helmholtz-Platz 1, 76344 Eggenstein-Leopoldshafen, Germany. [4] Institute of Organic Chemistry (IOC), Karlsruhe Institute of Technology (KIT), Fritz-Haber-Weg 6, 76131 Karlsruhe, Germany. [5] Institute of Toxicology and Genetics (ITG), Karlsruhe Institute of Technology (KIT), Hermann-von-Helmholtz-Platz 1, D-76344 Eggenstein-Leopoldshafen, Germany. * These authors contributed equally to this work. Correspondence and requests for materials should be addressed to A.K. (email: alexander.knebel@pci.uni-hannover.de) or to L.H. (email: Lars.Heinke@kit.edu).

Many sustainable applications require an efficient and energy-saving separation of molecular mixtures[1]. In this context, membrane technologies hold substantial potential for the separation of gas or liquid mixtures, since they offer an economic, energy-saving and ecological alternative to energy-intensive distillation or cryogenic separations. To reach high efficiencies, compelling research efforts are undertaken to develop novel materials allowing the fabrication of membranes with high permeation fluxes and high molecular separation factors. Ultrathin polymer[2] and porous carbon[3] membranes as well as two-dimensional materials, like graphene and graphene oxide[4,5], are considered to be auspicious materials in this context, since they offer high permeation fluxes, chemical stability, extensive mechanical strength and flexibility. However, a remaining major challenge is the precise control over the chemical functionality and over the pore size. Recently, metal-organic frameworks (MOFs)[6], crystalline solids composed of metal nodes connected by organic linker molecules, demonstrated large potential for membrane separation processes[7–12]. MOFs possess an abundance of unique properties like well-defined, periodic nanoporous structures, large specific surface areas and large structural and chemical variety, with more than 20,000 different MOF structures published before 2013 (ref. 6). Furthermore, the structure or chemical functionality of MOFs can be modified by different pre- and post-synthetic methods[13], which allow to further adjust the specific properties of these porous materials.

Of particular interest with respect to membrane fabrication is remote control, that is, the ability to switch crucial membrane parameters, like permeance or selectivity, by external stimuli without direct contact. Integrating such smart membranes into separation systems is expected to lead to a major break-through in overall performance. Light is a particularly simple, handy and (usually) non-invasive signal. Thus, photosensitive molecules, which undergo reversible isomerization when irradiated with light of a certain wavelength, attract a lot of attention[14–16]. Photoisomerization processes are typically very fast and

can be repeated many times. By incorporating photoswitchable molecular groups in molecular compounds like polymers and liquid crystals, some properties of these materials can be modified by illumination. Photoswitchable azobenzene-containing polymers and liquid crystals have been used as actuators[17,18] and light-driven motors[19]. Furthermore, photoswitchable molecular scaffolds[20] and gel assemblies[21] were realized utilising azobenzene photoisomerization. Also, for MOFs in the form of powders or thin films, photoswitchable moieties[22–24] were used to demonstrate the remote-control of the MOF properties like colour[25], release[26–28] and adsorption[29–33] of the guest molecules, in particular of $CO_2$ (refs 32,33). While in most cases illumination with ultraviolet-light was needed, linker molecules that can be switched with visible light were used for the MOF preparation recently[34].

Here, we describe the design of a nanoporous, photoswitchable MOF membrane, where the azobenzene moieties are incorporated as side groups into the framework. By controlling the isomerization state of the photosensitive groups by light, the flux and the separation factor of a molecular, gaseous mixture permeating the membrane can be switched. In addition to varying the permeability, the separation factor (also referred to as (perm-)selectivity) can be continuously tuned by adjusting the ratio of the *trans* and *cis* isomers of the photochromic moieties. As a result, the molecular composition of the permeate flux can be precisely tuned in a fully remote-controlled way, see Fig. 1.

## Results

**Photoswitchable SURMOF films.** The membranes are fabricated by growing MOF thin films in a layer-by-layer fashion, employing liquid-phase epitaxy by alternatively immersing the sample in solutions containing the MOF components, that is, in (separate) solutions of the metal nodes and of the organic linker molecules. This technique yields monolithic, crystalline, continuous MOF films, referred to as surface-mounted MOFs (SURMOFs)[35–38]. The prepared SURMOF films have a pillared-layer structure of type $Cu_2(AzoBPDC)_2(AzoBiPyB)$ (Fig. 2; AzoBPDC: 2-phenyldiazenyl-4,4′-biphenyldicarboxylic acid[26], and AzoBiPyB: (E)-4,4′-(2-(phenyldiazenyl)-1,4-phenylene) dipyridine (or dipyridylazobenzene)[30]. The samples used in this study were prepared by 90 (or 60) cycles of immersing the sample in ethanolic 1 mM copper(II)acetate solution and in ethanolic 0.1 mM AzoBiPyB and 0.1 mM AzoBPDC solution at 60 °C. Uptake experiments with butanol as guest molecule using a quartz crystal microbalance[39,40] indicate that the used MOF structure, that is, $Cu_2(AzoBPDC)_2(AzoBiPyB)$ with three azobenzene moieties per unit cell, results in large effects of the *trans-cis* switching, see Supplementary Fig. 1. The SURMOF films were prepared on quartz glass and on gold thin films on silicon supports for ultraviolet–vis (UV-vis) and infrared reflection absorption spectroscopy (IRRAS), respectively. For the membrane separation, the SURMOF films were prepared on asymmetric mesoporous aluminium oxide supports, which have a pore size of 70 nm on the top surface. Before the synthesis, the gold substrates were functionalized by 11-mercapto-1-undecanol (MUD) self-assembled monolayers (SAMs), while the quartz and porous aluminium oxide substrates were functionalized by oxygen plasma treatment[41].

From the IRRAS (Fig. 2b) and ultraviolet–vis spectra (Supplementary Fig. 2), it is clearly visible that the azobenzene (or phenyldiazenyl) side groups in the MOF can be switched by irradiation with light of 365 nm from *trans* (E) to *cis* (Z) and *vice versa* with light of 455 nm. The X-ray diffractograms of the SURMOF (Fig. 2c) indicate that the crystalline MOF (lattice) structure is not affected by the irradiation with ultraviolet or

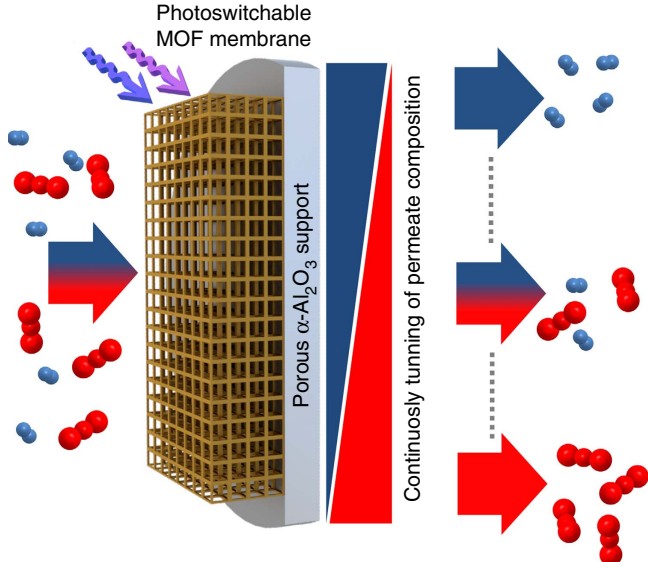

**Figure 1 | Switchable membrane separation.** Schematic illustration of tunable, remote-controllable molecular selectivity by a photoswitchable MOF membrane. The molecular feed mixture (left-hand side), where the molecules are depicted in red and blue, is separated by the nanoporous membrane. The molecular separation factor, giving the composition of the permeation flux (right-hand side), can be continuously tuned by light irradiation.

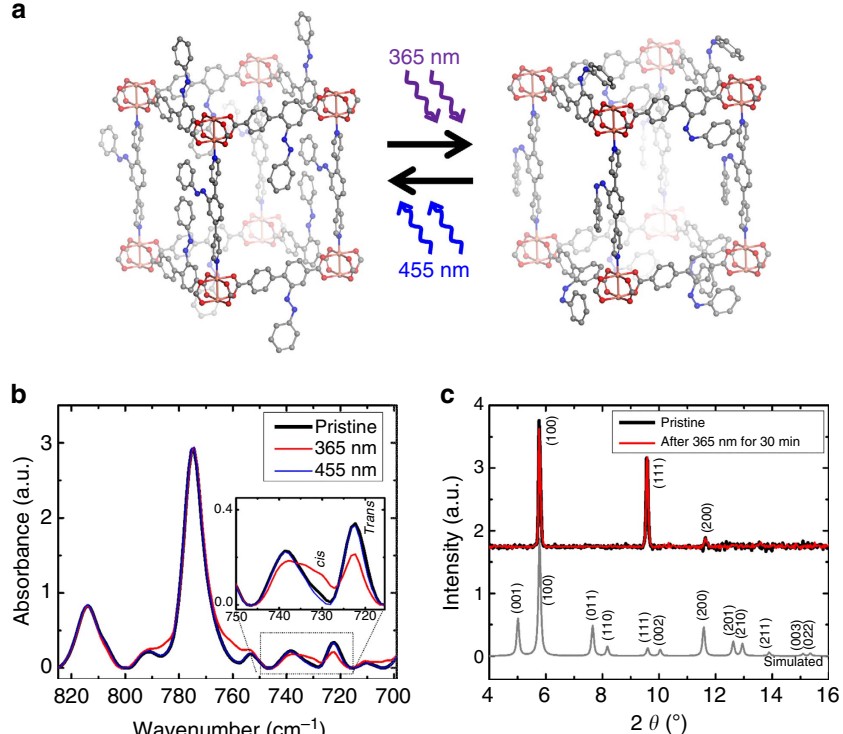

**Figure 2 | Photoswitchable SURMOFs of type Cu₂(AzoBPDC)₂(AzoBiPyB).** (**a**) The structure of Cu$_2$(AzoBPDC)$_2$(AzoBiPyB) with the azobenzene groups in the (basic) *trans* state, left, and in the *cis* state, right. The photoisomerization from *trans* to *cis* is performed by irradiation with ultraviolet-light of 365 nm, while the *cis*-to-*trans* isomerization is performed by irradiation with blue light of 455 nm. (Different view points of the MOF structure are shown in Supplementary Fig. 3.) (**b**) Infrared spectra of the Cu$_2$(AzoBPDC)$_2$(AzoBiPyB) SURMOF in the pristine *trans* state (black), after ultraviolet irradiation with light of 365 nm (red) and after irradiation with visible light of 455 nm (blue). The inset shows the magnification of the *trans* and *cis* azobenzene bands at about 720 and 730 cm$^{-1}$, respectively. An infrared spectrum with a larger range is shown in Supplementary Fig. 4. (**c**) Out-of-plane X-ray diffractogram of Cu$_2$(AzoBPDC)$_2$(AzoBiPyB) SURMOF before (black) and after (red) ultraviolet irradiation.

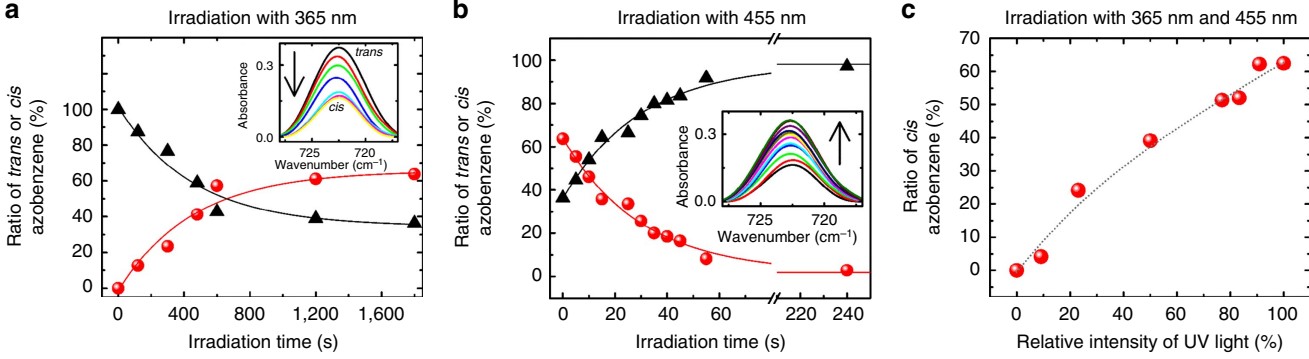

**Figure 3 | Photoswitching of the SURMOF films.** Amount of *trans* azobenzene (black triangles) and *cis* azobenzene (red spheres) of the Cu$_2$(AzoBPDC)$_2$(AzoBiPyB) SURMOF. (**a**) The sample is initially in the (100%) *trans* state and is irradiated with 365 nm. (**b**) The sample is initially in the *cis* state (that is, 63% *cis*) and is irradiated with 455 nm. The insets show the infrared *trans* azobenzene peak at 720 cm$^{-1}$ during the irradiation with 365 and 455 nm, respectively. (**c**) The sample is simultaneously irradiated with light of 365 and 455 nm. The irradiation time of each point is 30 min, resulting in a photostationary state. The ratio of *cis* azobenzene, determined from the infrared peak at 720 cm$^{-1}$, is shown as a function of intensity of 365 nm relative to the total light intensity (365 and 455 nm). The light intensities are (roughly) adjusted by the potentiometers of the LEDs.

visible light, this means, by the photoisomerization of the azobenzene-side-groups.

For a detailed understanding of the photoswitching in the MOF film, the photoisomerization during irradiation with ultraviolet or visible light or their mixture was studied by using IRRAS. By analysing the intensity of the infrared vibrational band at about 720 cm$^{-1}$, which can be assigned to *trans* azobenzene[42], the percentage of *trans* azobenzene and therefore of *cis*

azobenzene in the MOF structure was determined. This band may be assigned to the γ(CH) and τ(ring) vibration of the *trans* azobenzene side group, which, because of the bonding to the framework, is red shifted in comparison to the vibration band of the isolated azobenzene[43,44].

The maximum yield of *cis* azobenzene, this means, the photostationary state under irradiation by light of 365 nm wavelength, was determined to be ∼63%, Fig. 3a. The percentage of *cis* and

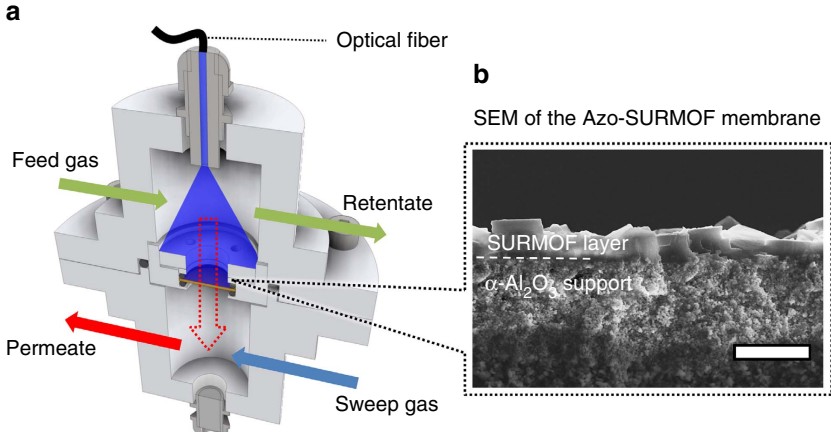

**Figure 4 | Membrane separation setup.** (**a**) Sketch of the setup for the photoswitchable membrane permeation experiments. The feed-retentate gas flow is above the membrane, while the sweep-permeate gas flow is below the membrane. A fiber-coupled LED is used for *in situ* irradiation of the SURMOF-covered top-side of the alumina membrane. (**b**) SEM cross-section image of the SURMOF membrane on the mesoporous α-alumina support. A dense SURMOF layer with a thickness of roughly 2 μm is observed. The length of the white scale bar is 5 μm.

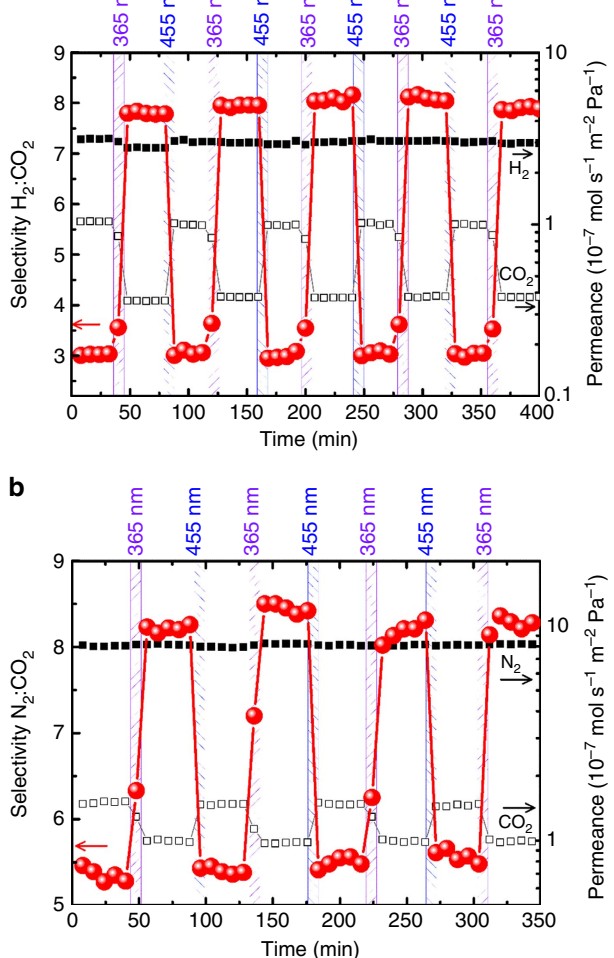

**Figure 5 | Photoswitchable membrane separation.** The separation of $H_2:CO_2$ and $N_2:CO_2$ mixtures is shown in (**a**,**b**), respectively. The membrane is irradiated by light of 365 and 455 nm for 5 min each. The permeances of $H_2$ and $CO_2$ (**a**), as well as $N_2$ and $CO_2$ (**b**) are shown as black solid squares and black open squares, respectively, with the logarithmic scale on the right-hand side. The molecular selectivities (or separation factors) are shown as red spheres with the scale on the left-hand side.

*trans* isomers can be adjusted by controlling the illumination time of the LED light with either 365 or 455 nm wavelength, see Fig. 3a,b. Moreover, a given *cis:trans* azobenzene ratio can also be achieved by simultaneously irradiating the SURMOF film with light of 365 and 455 nm, resulting in a steady state with equal rates of simultaneous *trans*-to-*cis* and *cis*-to-*trans* isomerizations. By adjusting the intensities of the respective LEDs with 365 or 455 nm wavelengths, a well-defined *cis:trans* ratio can be adjusted, see Fig. 3c. It should be noted that *cis* azobenzene can undergo thermal relaxation back to *trans* azobenzene with a thermal relaxation time of about 20 days at room temperature[42]. Therefore, the isomerization state of the photoswitchable SURMOF can be considered as stable in the dark, this means, without light irradiation, for the duration of the experiment (approximately a few hours).

**Tunable and remote-controllable membrane permeation.** The investigation of the photoswitchable membrane permeation was executed by growing $Cu_2(AzoBPDC)_2(AzoBiPyB)$ SURMOFs on mesoporous aluminium oxide supports. As shown by the SEM image in Fig. 4b, a continuous, dense and pinhole-free SURMOF film with a thickness of about 2 μm was formed on the mesoporous support. The X-ray diffractograms (Supplementary Fig. 5) verify the crystalline structure of the $Cu_2(AzoBPDC)_2$ (AzoBiPyB) SURMOF on the mesoporous support. For performing the membrane separation measurements, a Wicke–Kallenbach setup[45] with an optical fibre enabling the *in situ* irradiation of the MOF membrane was designed and built (Fig. 4a). This set-up allows to photoswitch the $Cu_2(AzoBPDC)_2(AzoBiPyB)$ SURMOF membrane, and to adjust the *cis:trans* azobenzene ratio during the membrane separation experiments. Here, the molecular separations of two different binary mixtures, $H_2:CO_2$ (pre-combustion mixtures) and $N_2:CO_2$ (post-combustion mixtures), were studied.

The membrane permeances and separation factors of hydrogen and carbon dioxide from the 50%:50% feed mixture are shown in Fig. 5a. The high separation factor indicates the absence of defects and pinholes in the membrane. While the permeation of hydrogen decreases only slightly, the permeation of carbon dioxide significantly decreases on switching the azobenzene-containing MOF membrane from *trans* to *cis*. The $CO_2$ permeance can be switched between ~4 and $10 \times 10^{-8}$ mol s$^{-1}$ m$^{-2}$ Pa$^{-1}$. The molecular $H_2:CO_2$ separation factor increases

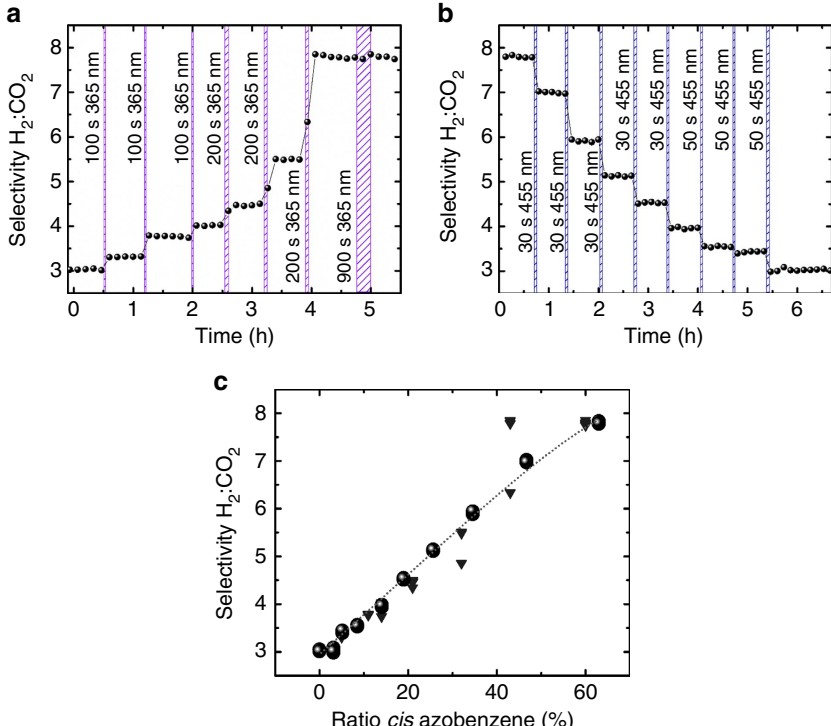

**Figure 6 | Tuning the H$_2$:CO$_2$ separation factor by adjusting the *cis*:*trans* azobenzene ratio.** (**a**) The Cu$_2$(AzoBPDC)$_2$(AzoBiPyB) membrane, which is initially in the *trans* state, is irradiated with ultraviolet-light for time intervals of 100 and 200 s, resulting in an increase of the H$_2$:CO$_2$ selectivity. (**b**) The membrane is initially in the *cis* state and is irradiated with 455 nm for time intervals of 30 and 50 s. (**c**) Correlation of the H$_2$:CO$_2$ separation factor with the ratio of *cis* azobenzene. The grey triangle and black spheres represent the data from figure (**a**,**b**), respectively; the correlation to the *cis* azobenzene ratio results from the comparison with the IRRAS data, see Supplementary Fig. 6.

from 3.0 in the *trans* state (100% *trans*, 0% *cis*) to about 8.0 in the *cis* state (37% *trans*, 63% *cis*). After irradiation with 455 nm, this means, the azobenzene-side-groups are switched back to the basic *trans* state, the initial permeances and the separation factor of 3 are attained. Five consecutive cycles of irradiation with 365 and 455 nm, this means, switching between the *trans* and the *cis* state, were performed. It was found that the switching of the membrane permeance and separation factor are fully reversible and, in addition, shows no sign of bleaching.

In a previous study with alkanes, alcohols and diols as guest molecules, it could be shown that the difference of the uptake by the azobenzene-containing MOF in the *cis* and in the *trans* state is caused by the switching of the azobenzene dipole moment, which amounts to 0 Debye in *trans* and 3 Debye in *cis*, and the dipole–dipole interaction with the guest molecules[39]. Thus, we may assume that the molecular mechanism resulting in the switching of the molecular separation is related to the different interaction of the hydrogen and carbon dioxide multipole moments with the *cis*- or *trans*-MOF, rather than a pore size effect or steric hindrance. Indeed, the dipole moment of the *cis* azobenzene is expected to result in a higher attractive interaction with the fairly strong quadrupole-moment of carbon dioxide, thus slowing down the carbon dioxide diffusion, while the interaction with hydrogen is not affected by the *trans*-*cis* switching. As a result, the different impact of the *trans*-*cis* switching leads to the switching of the H$_2$:CO$_2$ separation factor.

A reversible switching of the molecular selectivity is also observed for the separation of the N$_2$:CO$_2$ mixture, where the selectivity increases from 5.5 to 8.5 when switching from *trans* to *cis*, see Fig. 5b. While the carbon dioxide permeation significantly decreases on *trans*-to-*cis* isomerization, the nitrogen permeation is hardly affected. The observed permeation selectivity of nitrogen over carbon dioxide is characteristic for wide-pore

MOF/zeolite membranes with an adsorption controlled separation mechanism[46–48], whereas narrow-pore MOF/zeolite membranes show a preferred carbon dioxide (critical diameter 3.3 Å) permeation in comparison with nitrogen (3.6 Å) because of a diffusion-controlled molecular sieve mechanism[45,49,50].

A profound feature of the photoswitchable membrane is that the impressive (On–Off) switching of the separation factors (between membranes with minimum and maximum *cis* ratios) can be combined with the option of fine-tuning the *cis* azobenzene ratios. The H$_2$:CO$_2$ separation factor of the membrane in the *trans* state can be increased in arbitrary steps until the maximum separation factor is reached by irradiation with ultraviolet-light for certain time, Fig. 6a. On the other hand, the separation factor of the *cis* membrane can be decreased in arbitrary steps by irradiation with 455 nm, Fig. 6b. Thus, the separation factor can not only be switched between the minimum and maximum values, that is, 3.0 for 0% *cis* azobenzene and 8.0 for 63% *cis* azobenzene, but also each value in between can be realized by choosing appropriate irradiation times, that is, by adjusting the *cis* azobenzene ratio. The continuous tuning of the *cis* azobenzene ratio by controlled irradiation times with ultraviolet or visible light (Fig. 3) thus allows for a continuous tuning of the separation factor (Fig. 6). A correlation of the membrane selectivity with the *cis*-azobenzene ratio from the IRRAS data are shown in Fig. 6c. Because of the slow thermal isomerization of the azobenzene groups, the separation factor belonging to a mixed *cis*:*trans* state, once adjusted, is stable in the dark for the duration of the experiments.

## Discussion

The ratio of the azobenzene isomers can also be controlled by adjusting the relative intensities of the multi-component light

irradiation with ultraviolet and visible light (Fig. 3c). For instance, a $H_2$:$CO_2$ selectivity of 4.5 is reached for a *cis* ratio of about 20%. This ratio can be adjusted by irradiating the (100%) *trans* membrane with 365 nm for about 500 s (Fig. 6a), by irradiating the (63%) *cis* membrane with 455 nm for about 120 s (Fig. 6b) or by simultaneously irradiating the sample with 365 and 455 nm with an intensity ratio of about 20%:80% (Fig. 3c).

By increasing the *cis* ratio, that is, by increasing the irradiation time with ultraviolet-light or by increasing the relative ultraviolet intensity, the separation factor is increased. Here, the maximum ratio of *cis* azobenzene is about 63%, resulting in the highest separation factor of about 8.0. It can be assumed that even higher *cis*:*trans* isomerization ratios, for example, enabled by functionalized azobenzene moieties[51], would result in higher switching yields and would enlarge the tunable separation factor range.

The potential of the tunable membrane is demonstrated by controlling the molecular composition of the $H_2$:$CO_2$ permeate flux and, therewith, controlling the flammability and the safety of this pre-combustion mixture when mixed with air or oxygen. When the permeate flux is mixed with air in a ratio 1:20, the resulting gas flow through the membrane in the *trans* state has a hydrogen content of 3.7%, while the flux through the *cis* membrane has a hydrogen content of 4.4%. Since the flammability limit of hydrogen in air is 4.0% (ref. 52), the photoswitchable membrane in the *cis* state results in a flammable gas mixture and the *trans* membrane in a non-flammable gas mixture. When the permeate flux is mixed with air in a ratio 1:6, the flux through the *cis* membrane (with an $H_2$ content of 12.7%) may be detonable, while the flux through the *trans* membrane (with an $H_2$ content of 10.7%) is not detonable[52]. As demonstrated above, each composition in between can be tuned, enabling a remote-controlled adjusting of hazardous or non-hazardous conditions.

In conclusion, we have fabricated a novel MOF membrane, for which permeation, as well as selectivity can be remotely controlled by photoswitching the azobenzene-side-groups decorating the MOF pores. The separation factor of a $H_2$:$CO_2$ mixture was switched between 3 and 8. Moreover, the separation factor, and accordingly the molecular composition of the permeate flux, can be precisely adjusted within this range. This means for this pre-combustion gas mixture, the combustibility and ignition point (when mixed with air) can be continuously tuned by light irradiation of this smart membrane. The concept of switching and continuously tuning the membrane selectivities can be extended to the separation of other gas mixtures, for example, $N_2$:$CO_2$. Further innovative application might be the control of the accessibility to a catalyst or sensor surface and the controlled (selective) release of encapsulated fragrances or drugs. In addition, it could also be profoundly extended to other kinds of membranes like polymeric membranes[2] or mixed-matrix MOF membranes[9] by functionalizing these materials with photosensitive moieties.

## Methods

**X-ray diffraction.** X-ray diffraction measurements in out-of-plane geometry (also reffered to as co-planar orientation) were carried out using a Bruker D8-Advance diffractometer equipped with a position sensitive detector Lynxeye in $\theta$–$\theta$ geometry. Variable divergence slit and 2.3° Soller-slit were used on the secondary side. A Cu-anode with Cu $K_{\alpha 1}$ radiation ($\lambda = 0.154$ nm) was used. The measurements were carried out with a scan step of 0.02° at 40 kV and 40 mA.

**Infrared reflection-absorption spectroscopy.** IRRAS of the samples were performed with a resolution of 2 cm$^{-1}$ using a FTIR spectrometer of type Bruker VERTEX 80. All the IRRAS results were recorded in grazing incidence reflection mode at an angle of incidence of 80° relative to the surface normal using liquid-

nitrogen-cooled mercury–cadmium–telluride mid band detectors. Perdeuterated hexadecanethiol-SAMs on Au/Si were used for reference measurements.

**Ultraviolet–vis spectroscopy.** The ultraviolet–vis spectra were recorded by means of a Cary5000 spectrometer with a UMA unit from Agilent. Ultraviolet irradiation was performed with a 365 nm LED light from PrizMatix with a power or 112 mW. The visible light irradiation was performed with a 455 nm LED light with 135 mW. The distance between the sample and the LED was about 5 cm.

**Scanning electron microscopy.** SEM cross-section pictures were taken with a field-emission electron microscope JEOL-JSM-6700 F at 10 kV accelerating voltage and 10 μA emission current. To prevent the sample from charging, carbon was evaporated on the surface to provide a conductive surface coating using a Leica EMSCD500. The working distance was 15 mm.

**SAM-preparation.** For SAM formation, a clean gold substrate (that is, gold coated siliconwafer) was rinsed with pure ethanol and then immersed in a solution of MUD with a concentration of 1 mM in ethanol for 18 h. Afterwards the substrate was rinsed thoroughly with ethanol and dried under nitrogen stream.

**Plasma treatment.** The quartz glass and porous aluminium oxide substrates were treated by oxygen plasma (Diener Plasma; 50 sccm, pure $O_2$) for 10 min to remove impurities, as well as increase the number of OH functional groups and the hydrophilicy.

**SURMOF synthesis.** The experimental procedure used to grow SURMOF films on functional surfaces has been discussed in detail previously[35,37]. In summary, the layer-by-layer growth process consists of alternately immersing the substrate in the ethanolic solutions of the building units, that is, the metal knots (here: 1 mM copper acetate) and the organic linkers (here: 0.1 mM AzoBPDC and 0.1 mM AzoBiPyB; the synthesis of the linkers is explained in refs 26,30). Between each immersion step, the substrates were rinsed thoroughly with ethanol. In the present work, the substrates were immersed into copper acetate solution for 15 min, subsequently rinsed with pure ethanol solution for 2 min, and then immersed into the linker solution for 30 min. All solutions were kept at 60 °C during MOF film preparation. The SURMOF films on porous aluminium oxide supports (asymmetric mesoporous aluminium oxide supports from IKTS Hermsdorf, Germany) were prepared in 90 synthesis cycles. The SURMOF samples for infrared and ultraviolet–vis spectroscopy were prepared in 60 synthesis cycles on MUD-functionalized gold surfaces and on quartz glass, respectively. Since the sample is protected from light irradiation during the synthesis, all azobenzene-side-groups in the pristine SURMOF film are in the thermodynamically most stable state, the *trans* ground state.

**Gas permeation.** Gas permeation was carried out using an online gas chromatograph Agilent Technologies 7890A. Evaluation of the membrane was performed by mixed-gas permeation. Viton O-Rings (FKM 70 Vi 370) were used to seal the membrane gas-tight in its housing. *In-situ* irradiation of the membrane layer was achieved by a fibre-coupled, monochromatic high power Prizmatix FC-5 LED. Before the permeation data were recorded, the membrane was activated at 50 °C in a pure nitrogen flow for 24 h. For pre-combustion hydrogen separation at 25 °C, a 50:50 mixture of $H_2$:$CO_2$ was used at flow-rates of 25 ml min$^{-1}$ each. The feed-gas was $N_2$ at a 50 ml min$^{-1}$ flow rate. For post-combustion separation at 35 °C, a 90:10 $N_2$:$CO_2$ mixture was used at volumetric flow rates of 45 ml min$^{-1}$ for $N_2$ and 5 ml min$^{-1}$ for $CO_2$. There, $CH_4$ was applied on the sweep side at a flow rate of 50 ml min$^{-1}$. The experiments were performed at ambient pressure, that is, no overpressure was applied.

**Data availability.** The data sets generated during and/or analysed during the current study are available from the corresponding author on reasonable request.

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

## Acknowledgements

We gratefully acknowledge funding by the Volkswagen Foundation, by the BIFTM program of the Helmholtz Association and the German Science Foundation (DFG, SFB 1176, SPP 1928 and Ca147/20). Furthermore, we acknowledge support by DFG and Open Access Publishing Fund of Karlsruhe Institute of Technology.

## Author contributions

All authors contributed to writing the manuscript and have approved the final version of the manuscript. Z.W. synthesized the samples and performed the thin film switching experiments (X-ray diffraction, ultraviolet–vis, IRRAS). A.K. performed the membrane switching experiments and the SEM measurements. S.G., D.W. and S.B. synthesized the photoswitchable molecules. C.W. and J.C. were involved in planning and supervising the experiments. L.H. planned and supervised the experiments and wrote the manuscript.

## Additional information

**Competing financial interests:** The authors declare no competing financial interests.

**Publisher's note**: 

