## [Peer Review File · Nature Communications]

Reviewers' comments:

Reviewer #1 (Remarks to the Author):

The author reported an azobenzene-based MOF membrane, in which the permeation as well as selectivity can be remotely controlled by light irradiation. The separation factor of a H₂:CO₂ mixture can be switched between 3 and 8, and also can be precisely adjusted within this range. The work is interesting (even the separation factor is not so impressive) and generally speaking, it has been carried out well. Given the important research interest in stimuli-responsive MOFs and also MOF membrane, I would like to recommend the publication of this manuscript on Nature Communications after addressing the questions stated below:

1. The authors claimed the initial state of the azobenzenes are all in trans state (100% trans) in the acquired MOF membrane. Do they have any evidences? Usually, both trans and cis should be existing in the initial state. The author need to reconsider about this conclusion.
2. The author assume that the molecular separation is related to the different interaction of the hydrogen and carbon dioxide multipole moments with the cis- or trans-MOF rather than a pore size change. Have they investigated the gas adsorption and pore size change before and after UV irradiation? Please include the pore size calculations of both cis and trans pores.
3. Figure 1 is a little bit confused, especially both the gas molecules and arrow are colored in blue and red.
4. Several references about stimuli-responsive MOFs can be added: Chem Asian J. 2014, 9, 2358. Sci. Adv. 2016, 2, e1600480; Chem. Mater., 2015, 27, 6426.

Reviewer #2 (Remarks to the Author):

This work shows the first example of membrane that switches CO₂ permeability with light. The system consists on a Cu-based MOF grown layer-by-layer on a monolithic mesoporous alumina support. The authors show how by shining 365 nm UV light on the MOF/Alumina membrane, the CO₂ permeability in both H₂:CO₂ and N₂:CO₂ mixtures decreases, whereas by subsequently exposing the MOF to 455 nm light, CO₂ permeability goes back to its initial value. In this way the separation index can be reversibly switched between 3.0 and 8.0 and 5.5 and 8.5 for H₂:CO₂ and N₂:CO₂ mixtures, respectively. The degree of trans to cis isomerization in the MOF is estimated from IR spectroscopy (IRRAS). The changes in adsorption properties are attributed the different dipole moments of the trans and cis configuration of the azobenzene pending groups, so that the CO₂ permeability can be tuned by controlling the trans/cis ratio in the MOF.

I recommend the publication in Nature Communications. The results are novel and an important contribution to the field because it implements a photoswitchable MOF in a real application. The thin film MOF on alumina support is a smart way of creating a membrane where light penetration is not an issue and a large area can be coated. The system seems to be robust and yields reproducible photoswitching.

However, a point that could be better explained is the estimation of trans/cis ratio. The authors claim that the signal at 720 cm⁻¹ in the IRRAS spectrum arises from g(CH) and t(ring) vibrations, according to Duarte's publication. This estimation was already used by some of the authors in a previous paper (Phys. Chem. Chem. Phys. 2015). However, in Duarte's work, such vibrations for the E (trans) configuration appear at 780 cm⁻¹ and for the Z (cis) configuration at 776 cm⁻¹. The authors could comment on how did they conclude that the band at 720 cm⁻¹ arises only from the E form of the azobenzene and it does not have any contribution from the Z form, given the closeness of the vibration modes of both configurations in Duarte's studies.

Another question that should be clearer is if the IRRAS experiments performed on MOF layers' on

Si were the same thickness as in the alumina membrane. It was not obvious in the paper and it is important to know that the systems were analog since IRRAS was used to quantify the trans/cis ratios that render different CO₂ permeability.

It would be very informative to have a broader window in the IR spectrum to have full evidence of pureness of the MOF (for example that there are no free linkers trapped in the pores). This info could just be in the supporting material.

In page 2 line 41 "a major challenge remains to be" does not sound correct and could be substituted by "a major challenge remaining is".

The format of the references is not homogeneous.

Reviewer #3 (Remarks to the Author):

Heinke et al report the synthesis of a photoresponsive SURMOF layer. They demonstrate that they are able to switch the porosity within the membrane layer, and that this results in a significant change in the gas selectivity within the membrane.

This is a great paper and of strong interest to the community. the study has been conducted very thoroughly and I dont see the need to conduct extra experiments.

I recommend that it be published with minor revisions.

The only thing that I would recommend the authors reexamine is the significance of what they have discovered. While of lot of the introduction and discussion canvases the use of the membrane as being adaptable to gas composition, in my view this is not the true area where this discovery will find application.

Gas separation membranes commonly suffer from concentration polarisation due to a build up of the slowly permeating gas at the membrane surface. Richard Baker in his many books and review article commonly discusses this for polymers. By being able to switch the selectivity using light, the membrane can be 'reactivated'. I recommend the authors consider the significance of their work in terms of offsetting concentration polarisation in light of the literature on this topic, as it is a true breakthrough that can change how gas separation membrane operate.

Reviewer #1

The author reported an azobenzene-based MOF membrane, in which the permeation as well as selectivity can be remotely controlled by light irradiation. The separation factor of a H₂:CO₂ mixture can be switched between 3 and 8, and also can be precisely adjusted within this range. The work is interesting (even the separation factor is not so impressive) and generally speaking, it has been carried out well. Given the important research interest in stimuli-responsive MOFs and also MOF membrane, I would like to recommend the publication of this manuscript on Nature Communications after addressing the questions stated below:

1. The authors claimed the initial state of the azobenzenes are all in trans state (100% trans) in the acquired MOF membrane. Do they have any evidences? Usually, both trans and cis should be existing in the initial state. The author need to reconsider about this conclusion.

Response: We gratefully acknowledge the positive comments from reviewer #1. Our statement that the azobenzene-SURMOF is initially (i.e. after the synthesis) in the trans state is based on the fact that the trans state is thermodynamically more stable ground state of azobenzene. At room temperature, after excitation to the cis-state with UV light, azobenzene relaxes back to the trans state within a few hours or days. (The trans isomer is more stable by approximately 50 kJ/mol, and the barrier to isomerization in the ground state is approximately 100 kJ/mol. see DOI: 10.1039/C1CS15179G)

To stress this point, we added to the text: "Since the sample is protected from light irradiation during the synthesis, all azobenzene side groups in the pristine SURMOF film are in the thermodynamically most stable state, the trans ground state."

An unambiguous proof of the azobenzene side groups in the pristine SURMOF adopting the trans (ground) state is provided by the IRRAS shown in Figure 2b, as well as by the UV-vis spectra reproduced in Fig. S15.

2. The author assume that the molecular separation is related to the different interaction of the hydrogen and carbon dioxide multipole moments with the cis- or trans-MOF rather than a pore size change. Have they investigated the gas adsorption and pore size change before and after UV irradiation? Please include the pore size calculations of both cis and trans pores.

Response: We agree with the reviewer that the switching of azobenzene from *trans* state to *cis* state not only induces the change of the dipole moment, but also modifies the framework, the pore size and pore structure.

Single-component gas adsorption experiments with azobenzene-containing SURMOF films were performed in ref. 25 and 29. In DOI: 10.1002/cphc.201500829, it was found that the effect of the photoswitching on the adsorption can be explained by dipole-dipole (Keesom) interaction.

We followed the reviewer's suggestion and calculated (optimized by MaterialStudio) the pore structure of the *trans* and *cis* SURMOF. This is added as supporting information S18. The added figure caption is: "Comparison of the Cu₂(AzoBPDC)₂(AzoBiPyB) SURMOF structure with the azobenzene side groups in the *trans* and the *cis* state. It should be noted that the phenyl rings, and thus the azobenzene side groups, can rotate around the bond axis. Therefore, the positions of the azobenzene side groups are not fixed (or "frozen") as depicted in Figure S18."

3. Figure 1 is a little bit confused, especially both the gas molecules and arrow are colored in blue and red.

Response: The colors (red and blue) of the arrows represent the permeation possibility of the gas molecules (red and blue), demonstrating the continuously tuning of the permeation flux. We improved Figure 1 (added the “feed” and “permeate” labels) and the figure caption (added the sentence: “The molecular feed mixture, where the molecules are depicted in red and blue, is separated by the nanoporous membrane.”).

4. Several references about stimuli-responsive MOFs can be added: *Chem Asian J.* 2014, 9, 2358. *Sci. Adv.* 2016, 2, e1600480; *Chem. Mater.*, 2015, 27, 6426.

Response: We have followed the suggestion of the reviewer and have added the references suggested by the referee (*Chem Asian J.* 2014, 9, 2358. *Sci. Adv.* 2016, 2, e1600480). In addition, we added a citation of DOI: 10.1039/C5CE02543E.

Reviewer #2

This work shows the first example of membrane that switches CO₂ permeability with light. The system consists on a Cu-based MOF grown layer-by-layer on a monolithic mesoporous alumina support. The authors show how by shining 365 nm UV light on the MOF/Alumina membrane, the CO₂ permeability in both H₂:CO₂ and N₂:CO₂ mixtures decreases, whereas by subsequently exposing the MOF to 455 nm light, CO₂ permeability goes back to its initial value. In this way the separation index can be reversibly switched between 3.0 and 8.0 and 5.5 and 8.5 for H₂:CO₂ and N₂:CO₂ mixtures, respectively. The degree of trans to cis isomerization in the MOF is estimated from IR spectroscopy (IRRAS). The changes in adsorption properties are attributed the different dipole moments of the trans and cis configuration of the azobenzene pending groups, so that the CO₂ permeability can be tuned by controlling the trans/cis ratio in the MOF.

I recommend the publication in Nature Communications. The results are novel and an important contribution to the field because it implements a photoswitchable MOF in a real application. The thin film MOF on alumina support is a smart way of creating a membrane where light penetration is not an issue and a large area can be coated. The system seems to be robust and yields reproducible photoswitching.

*However, a point that could be better explained is the estimation of trans/cis ratio. The authors claim that the signal at 720 cm⁻¹ in the IRRAS spectrum arises from g(CH) and t(ring) vibrations, according to Duarte's publication. This estimation was already used by some of the authors in a previous paper (*Phys. Chem. Chem. Phys.* 2015). However, in Duarte's work, such vibrations for the E (trans) configuration appear at 780 cm⁻¹ and for the Z (cis) configuration at 776 cm⁻¹. The authors could comment on how did they conclude that the band at 720 cm⁻¹ arises only from the E form of the azobenzene and it does not have any contribution from the Z form, given the closeness of the vibration modes of both configurations in Duarte's studies.*

Response: We also gratefully acknowledge the positive comments from reviewer 2.

The intensity of the IR absorption band at about 720 cm⁻¹ decreases upon irradiation with UV light (i.e. trans-to-cis switching as seen by UV-vis) and increases again upon irradiation with visible light or by thermal relaxation. Accordingly, the corresponding vibration must be specific to the *trans* azobenzene conformation. The IR bands at about 730 cm⁻¹, on the other hand, increase after UV irradiation and are thus related to a vibration of *cis* azobenzene. This *cis* band was also found by Ruschewitz et. al. (dx.doi.org/10.1021/ic302856b)

Based on the work by Duarte *et al.*, (DOI: 10.1039/c4cp00240g) where isolated azobenzene molecules in an argon matrix were studied combined with DFT calculations, the band at 720 cm⁻¹ could be assigned

to the $\gamma(\text{CH})$ and $\tau(\text{ring})$ vibrations of *trans* azobenzene and the band at 705 cm^{-1} to the corresponding vibration of *cis* azobenzene. We assume that the red shift relative to the frequencies reported by Duarte et al. results from the differences in the molecular structure, i.e. from attaching the azobenzene moieties to the MOF framework.

We followed the reviewer suggestion and explain this point in the manuscript:
“By analyzing the intensity of the IR vibrational band at about 720 cm^{-1} , which can be assigned to *trans* azobenzene,³⁹ the percentage of *trans* azobenzene and therefore of *cis* azobenzene in the MOF structure was determined. This band might be assigned to the $\gamma(\text{CH})$ and $\tau(\text{ring})$ vibration of the *trans* azobenzene side group, which, due to the bonding to the framework, is red shifted in comparison to the vibration band of the isolated azobenzene.^{40,41}”

Another question that should be clearer is if the IRRAS experiments performed on MOF layers' on Si were the same thickness as in the alumina membrane. It was not obvious in the paper and it is important to know that the systems were analog since IRRAS was used to quantify the trans/cis ratios that render different CO2 permeability.

Response: We thank the reviewer for this comment. In order to make this point more clear, we have added the following sentences to the manuscript: “The SURMOF films on porous aluminum oxide supports were prepared in 90 synthesis cycles. The SURMOF samples for IR and UV-vis spectroscopy were prepared in 60 synthesis cycles on MUD-functionalized gold surfaces and on quartz glass, respectively.”

For the IRRAS and UV-vis experiments, we used SURMOF films with a thickness that enable ideal spectra (i.e. with small signal-to-noise ratios). These SURMOF films are thinner than the SURMOF membrane. In this study we consider the switching of the trans:cis ratio, which is independent of the film thickness.

It would be very informative to have a broader window in the IR spectrum to have full evidence of pureness of the MOF (for example that there are no free linkers trapped in the pores). This info could just be in the supporting material.

Response: In following the reviewer's suggestion, the revised version of the supporting information now contains an IR spectrum with a wider range (Figure SI9).

A very small vibrational band at around 1705 cm^{-1} is present, which is assigned to the C-O double band in the carboxyl acid group of the organic linker. This observation suggests that the number of free linkers trapped in the pores must be very small. Note, that a small band arising from free carboxylic acid groups has to be expected, since the last layer grown in the SURMOF-process is carried out by immersion in a solution of the organic linkers.

In page 2 line 41 “a major challenge remains to be” does not sound correct and could be substituted by “a major challenge remaining is”.

Response: We followed the suggestion and changed the text to: “However, a remaining major challenge is the precise control over the chemical functionality and over the pore size.”

The format of the references is not homogeneous.

Response: We have changed the format of the references according to the regulation of Nature Communications.

Reviewer #3

Heinke et al report the synthesis of a photoresponsive SURMOF layer. They demonstrate that they are able to switch the porosity within the membrane layer, and that this results in a significant change in the gas selectivity within the membrane.

This is a great paper and of strong interest to the community. the study has been conducted very thoroughly and I dont see the need to conduct extra experiments.

I recommend that it be published with minor revisions.

The only thing that I would recommend the authors reexamine is the significance of what they have discovered. While of lot of the introduction and discussion canvases the use of the membrane as being adaptable to gas composition, in my view this is not the true area where this discovery will find application.

Gas separation membranes commonly suffer from concentration polarisation due to a build up of the slowly permeating gas at the membrane surface. Richard Baker in his many books and review article commonly discusses this for polymers. By being able to switch the selectivity using light, the membrane can be 'reactivated'. I recommend the authors consider the significance of their work in terms of offsetting concentration polarisation in light of the literature on this topic, as it is a true breakthrough that can change how gas separation membrane operate.

Response: We are delight to learn about the very positive opinion of this referee. We have followed all recommendations of this expert in the field and have added further potential applications. The new version of the manuscript now contains the following sentences: "Further innovative application might be the control of the accessibility to a catalyst or sensor surface and the controlled (selective) release of encapsulated fragrances or drugs." Although we would like to use this photoswitchable membrane also to overcome the concentration polarization in membranes, we did not find a suitable setup or experiment where this photoswitchable membrane would hinder the concentration polarization.

REVIEWERS' COMMENTS:

Reviewer #1 (Remarks to the Author):

Since the authors have addressed most of the comments properly, the manuscript could be recommended for publication in nature communications.

Reviewer #2 (Remarks to the Author):

The authors of this paper addressed the reviewers comments fully. The manuscript improved with the info added in the new version, according to the reviewers' comments.

Just as a final remark about one of the author's answers. It is true that the trans/cis ratio is independent of the film thickness in the photostationary state. But film thickness affects light penetration and thus the time needed to reach the PSS is usually different for different thickness.

I recommend the publication of this manuscript in Nature Communications without further changes.